# The Effect of Fin Shape on the Heat Transfer and the Solution Time of a Microchannel Evaporator in a CO_2_ Air Conditioning System—A Numerical Investigation

**DOI:** 10.3390/mi13101648

**Published:** 2022-09-30

**Authors:** Tronghieu Nguyen, Thanhtrung Dang

**Affiliations:** Department of Thermal Engineering, HCMC University of Technology and Education, Ho Chi Minh City 71307, Vietnam

**Keywords:** fin shape, CO_2_, microchannel evaporator, heat flux, temperature profile

## Abstract

Numerical simulations on the fin shape of a microchannel evaporator in a CO_2_ air conditioning system were performed at the inlet evaporative temperature of 10 °C and the vapor quality of 0.61. Two types of fin shapes were dealt with: the straight fins and V-fins. The numerical results were verified by the experimental data. For the system under consideration and for the same heat transfer area and the heat transfer coefficient for the air side in the microchannel evaporator, the effect of the fin shape on the heat transfer was not different; however, the solution time and the physical memory for the straight fins were 1.3 and 1.45 times compared with the V-fins, respectively. Therefore, the V-fin shape should be used for numerical simulation to compare it with the straight fin shape. In this study, the evaporation of the refrigerant in the microchannel evaporator took place in four passes. The normal heat flux from the air through the fins and tubes was almost reached at 1550 W/m^2^ at the evaporative temperature of 10 °C. The results obtained from the experimental data were in good agreement with those obtained from the numerical results, with a deviation of less than 10%.

## 1. Introduction

Heat exchangers, especially fin tube heat exchangers, are widely used in industrial and domestic applications, such as air conditioners, heat pumps, refrigeration systems, cooling systems, etc. In the fin tube heat exchangers, the geometries of the fin, including the pitch, thickness, shape and angle, are important parameters, which strongly affect their capacities. Recently, many types of research related to the effect of the fin shape on the performance of the exchanger have been reported [1,2,3,4,5,6]. For instance, Hsieh et al. [5] optimized the louver finned-tube heat exchangers using the Taguchi method. The results indicated that three main factors, including fin collar outside diameter, transverse tube pitch and fin pitch, significantly influence the thermal hydraulic performance of the heat exchanger [5].

Recently, the application of CO_2_ as a refrigerant in the fin tube heat exchangers has attracted particular attention due to its non-toxicity, safety and environmental friendliness. Unfortunately, the previous report showed that the use of CO_2_ in an air conditioning system led to its low performance [7]. Thus, to increase the efficiency of heat exchange in the CO_2_ systems and reduce their size as well, lately, the microchannel or minichannel heat exchangers have been installed in the systems [8,9,10,11]. Moreover, to develop the new CO_2_ microchannel system in a short time and at a lower cost, the numerical simulation has been extensively used [9,12,13,14,15]. For example, based on the finite volume method, Jin et al. [11] developed and predicted the performance of a CO_2_ transcritical system. The model had a mean deviation with the experimental cooling capacities of 1.9% and with refrigerant-side pressure drops of 12.3%. Using numerical analysis, Yun et al. [12] developed a microchannel evaporator for a CO_2_ air conditioning system. The performance of the developed model can be improved by changing the mass flow rate and varying the fin pitch to expand the two-phase region. Furthermore, also applying the finite volume method, Kim and Bullard [13] developed a two-slab microchannel evaporator for a CO_2_ air conditioning system. However, the results in Refs [7,8,9,10,11,12,13,14,15] did not mention the solution time for numerical simulation in the microchannel evaporator.

To verify the theoretical calculation and the numerical simulation or to find the physical rules for the models, the corresponding experimental results are needed for developing the fin tube heat exchangers. Patino et al. [16] used experimental data to demonstrate heat transfer correlations in a CO_2_ evaporator model, which was developed using a finite volume method. For both numerical and experimental methods, Yang and Ning [17] studied a CO_2_ double pipe evaporator to find out the effects of the tube diameters and the operation parameters on its performance. The experimental data yielded a good correlation with the evaporator heat transfer and pressure drop, verified by the relative errors of 5.21% and 3.78%, respectively.

From the literature above, it is found that numerical simulation is a strong method for developing a fin tube microchannel evaporator using CO_2_ as a refrigerant. Moreover, the corresponding experiments are needed to enhance the system’s reliability. The review results did not deal with the fin shape and solution time in more detail. Herein, we numerically investigate the effect of the fin shape and the solution time on the heat transfer of a microchannel evaporator in a CO_2_ air conditioning system. There are two types of fin shapes: V-shaped fins and straight fins. Moreover, experiments are also carried out to demonstrate the numerical simulation results.

## 2. Methodology

### 2.1. Design of Evaporator Model

The flat tube evaporator was designed with a heat transfer area of 2.5 m^2^ and a cooling capacity of 2.6 kW. Figure 1 displays the dimensions of the microchannel evaporator. The evaporator is composed of 29 flat tubes and divided into 6 passes. There are 3, 4, 5, 6, 6 and 5 flat tubes in the first, second, third, fourth, fifth and sixth pass, respectively. Each flat tube has 10 rectangular microchannels with dimensions of 1.2 mm × 0.6 mm.

There are two fin shapes being investigated with the same heat transfer area and the heat transfer coefficient for the air side. The straight fins have a height of 4.1 mm, and the V-fins have a height of 4.05 mm, as shown in Figure 1. Table 1 presents the detailed evaporator specifications.

### 2.2. Mathematical Model

In this study, a turbulent model was used for numerical simulation. The governing equations are shown as follows [18].

The momentum formulation is as follows:(1)$$\rho u\cdot \nabla \left(u\right)=\nabla \cdot \left[-p+\left(\mu +\mu_{T}\right)\left(\nabla u+{\left(\nabla u\right)}^{T}\right)-\frac{2}{3}\left(\mu +\mu_{T}\right)\left(\nabla \cdot u\right)I\right]+F$$

The conservation of mass equation
(2)∇·(ρu)=0

Heat transfer equations
(3)$${\rho C}_{p}u\cdot \nabla T+\nabla \cdot q=Q$$
(4)q=−k∇T
where T is the absolute temperature (K), p is the pressure (Pa), **u** is the fluid velocity (m/s), $\mu_{T}$ is the turbulence viscosity (Pa.s), F is the external forces applied to the fluid (N/m^3^), ρ is the density (kg/m^3^), Cp is the specific heat capacity at constant pressure (J/(kg·K)), Q contains additional heat sources (W/m^3^), and q is the heat flux (W/m^2^).

The phase change equations are as follows:(5)$$\rho =\theta_{1}\rho_{1}+\theta_{2}\rho_{2}$$

The specific heat capacity is
(6)$$C_{p}=\frac{1}{\rho }\left(\theta_{1}\rho_{1}C_{p,1}+\theta_{2}\rho_{2}C_{p,2}\right)+L_{1-2}\frac{\partial \alpha_{m}}{\partial T}$$

The mass fraction is
(7)$$\alpha_{m}=\frac{1}{2}\frac{\theta_{2}\rho_{2}-\theta_{1}\rho_{1}}{\theta_{1}\rho_{1}+\theta_{2}\rho_{2}}$$

The effective thermal conductivity is as follows:(8)$$k=\theta_{1}k_{1}+\theta_{2}k_{2}$$

The velocity of the refrigerant in the microchannel is
(9)$$u_{\mathrm{ch}}=\frac{\dot{m}}{{n\rho A}_{\mathrm{ch}}}$$

The relationship between the vapor quality and the phase indicator is
(10)$$x=\theta_{2}\;\mathrm{and}\;1-x=\theta_{1}$$
(11)$$\theta_{2}=\frac{h-h_{1}}{L_{1-2}}$$
where indices 1 and 2 indicate a refrigerant in phase 1 (liquid) and in phase 2 (vapor), respectively, h is the enthalpy (kJ/kg), L_1-2_ is the latent heat (J/kg), x is the vapor quality, and θ is the phase indicator.

In addition, a general k—ε model was selected for the turbulent flow in the present simulation. The equations of the k—ε model are shown below.

The turbulent viscosity is modeled as
(12)$$\mu_{T}={\rho C}_{\mu }\frac{k_{T}^{2}}{\varepsilon }$$

The transport equation for k reads
(13)$$\rho \left(u\cdot \nabla \right)k=\nabla \cdot \left[\left(\mu +\frac{\mu_{T}}{\sigma_{k}}\right)\nabla k\right]+p_{k}-\rho \varepsilon$$
where the production term $p_{k}$ is
(14)$$p_{k}=\mu_{T}\left[\nabla u∶\left(\nabla u+{\left(\nabla u\right)}^{T}\right)-\frac{2}{3}{\left(\nabla \cdot u\right)}^{2}\right]-\frac{2}{3}\rho_{k}\nabla \cdot u$$

The transport equation for ε is as follows: (15)$$\rho \left(u\cdot \nabla \right)\varepsilon =\nabla \cdot \left[\left(\mu +\frac{\mu_{T}}{\sigma_{\varepsilon }}\right)\nabla \varepsilon \right]+C_{\varepsilon 1}\frac{\varepsilon }{k_{T}}p_{k}-C_{\varepsilon 2}\frac{\varepsilon^{2}}{k_{T}}$$
where $k_{T}$ is the turbulent viscosity. For the superheated passes, the phase indicator θ_1_ (liquid) equals zero. The steady-state method was used to solve the CO_2_ microchannel evaporator.

### 2.3. Numerical Simulation

It is very difficult and complex to simulate the microchannel evaporator at one time because the evaporator has thin fins and microchannels. Therefore, there are six simulation times corresponding to six passes of the evaporator. This approach is used to reduce the computing resources. The input conditions for the first pass are the parameters in Table 2. The input conditions for the second pass are the numerical simulation results of the first pass and so on, until the numerical simulation of the sixth pass is completed. Table 2 shows some inlet parameters and outlet parameters. The inlet parameters include some initial values. The unknown parameters were calculated using Engineering Equation Solver (EES) [19], REFPROP [20] and DORIN [21] software.

Figure 2 shows two flat tube models of the CO_2_ microchannel evaporator. The CO_2_ flows inside the flat tube, and the air is outside with the convective heat transfer. To determine the variations of the vapor quality at the evaporator inlet, the EES software [19] was used based on some initial conditions and the pressure–enthalpy diagram of R744.

In Figure 3, the finite elements in the grid meshes of the straight fin model were partitioned to be triangular. For the fluid flow, the boundary layer was divided into 5 layers to investigate the characteristics of the wall, as shown in Figure 3a. For important parts, the finer mesh level was used, i.e., the edges of the corners in the microchannels were surveyed on 4 different levels (1, 10, 20, 30), as shown in Figure 3b. For the V-fin model, the method and the meshing level are the same as in the straight fin model. The model consists of approximately 610,000 domain elements and 122,000 boundary elements. The average number of degrees of freedom (DOF) was 11,300,000 (plus 134,000 internal DOFs). The parallel sparse direct solver (PARDISO) algorithm is a solver that was developed based on the linear equation system. The relative tolerance was set at 10^−6^. For numerical simulation in this study, the COMSOL Multiphysics software-version 6.0 was used with the PARDISO solver.

### 2.4. Experimental Setup

Figure 4 shows a schematic diagram of the test apparatus used in the study. It consists of a CO_2_ DORIN compressor, a throttle valve, a gas cooler, a V-fin tube evaporator and an internal heat exchanger. The pressure sensors and the thermocouples were installed at the inlet and the outlet of each component to measure the pressure and temperature. Their ranges and accuracies of measurement are displayed in Table 3. The mass flow rate of the refrigerant was measured by a digital volumetric flow rate meter or determined by the DORIN software based on the suction temperature, the evaporative temperature and the gas cooler pressure. The experimental data of the microchannel evaporator with a V-finned flat tube will be used to validate the numerical simulation results for the flat tube with straight fins and V-fins. From Figure 5, the experimental data were imported to the EES software; the vapor quality was determined by x = 0.61; the value was used to import the numerical simulation. The uncertainty values of several parameters, such as the vapor quality and the normal heat flux, were 3.17% and 6.2%, respectively. The evaporator with the V-finned flat tube for the CO_2_ air conditioning system is shown in Figure 6.

## 3. Results and Discussion

The experimental data obtained from the test loop were under the ambient (outdoor) temperature of 33 °C. The inlet evaporative temperature and inlet vapor quality were fixed at 10 °C and 0.61, respectively. The temperature slice of the straight finned flat tube is shown in Figure 7. The inlet and outlet refrigeration temperatures of the straight finned flat tube were the same in the first pass of the microchannel evaporator; this result is consistent with the theory of evaporation. The temperature of the fins was about 1 °C higher than that of the CO_2_ refrigerant.

With the same initial simulation conditions, the temperature slice on the V-finned flat tube was the same as that obtained from the straight finned flat tube, as shown in Figure 8. The temperature profile obtained from the straight finned flat tube was also the same as that obtained from the V-finned flat tube, as shown in Figure 9 and Figure 10. However, the arrows of the heat fluxes were different for these two types. The conductive heat flux of the V-finned flat tube model was more concentrated in the connection location between the fins and the flat tube.

Figure 11 shows the vapor quality of the straight finned flat tube evaporator model and the V-finned flat tube evaporator model at the evaporative temperature of 10 °C. Due to the same heat transfer area (2.5 m^2^), the straight fins had a height of 4.1 mm, and the V-shaped fins had a height of 4.05 mm (Figure 1). Therefore, the simulated vapor qualities in these two cases were the same. Each microchannel pass of the evaporator had a length of 300 mm. The evaporation of the refrigerant took place in four passes, so the normal heat flux from the air through the fins and tubes was almost reached at 1550 W/m^2^. It was observed that the saturated vapor state (dry state) was achieved at the outlet of the fourth pass. When the CO_2_ was superheated, the normal heat flux rapidly declined from 1550 to 700 W/m^2^. From Figure 7, Figure 8, Figure 9, Figure 10 and Figure 11, with the same heat transfer area and the heat transfer coefficient for the air side, the effect of the fin shape on the heat transfer of a microchannel evaporator was no different between the straight fin model and the V-fin model.

Figure 12 shows the velocity field of the refrigerant along the length of the flat tube at the evaporative temperature of 10 °C. At the inlet, the velocity rose sharply because the refrigerant from the manifold entered the microchannel with a smaller cross-section. The velocity increased gradually in the microchannel; this was due to the mixture density decreasing. At the outlet, the velocity fell rapidly due to the refrigerant leaving the microchannel to enter the manifold, which saw the cross-section increase suddenly.

The comparison between the numerical simulation results and the experimental data is shown in Figure 13 at the evaporative temperature of 10 °C. The experimental temperature profile was captured by an infrared thermal camera and then compared with the numerical simulation results. The experimental temperatures were also obtained from the thermocouples. It could be observed that they had similarities at the first four passes, and the refrigerant was superheated at the fifth and sixth passes. From the figure, the deviation between the numerical results and the experimental data was less than 10% at the hottest spot.

Table 4 shows the results on the number of elements, the solution time and the physical memory between the straight finned flat tube model and the V-finned flat tube model. For the same method and meshing level, the V-fin tube model had more elements but less solution time and physical memory compared with those obtained from the straight fin tube model. The results showed that the fin shape affected the solution time and the memory. The solution time and the physical memory for the straight fins were 1.3 and 1.45 times those of the V-fins, respectively. This could be due to the grid meshes and the PARDISO solver in the V-fin tube model being more suitable than those obtained from the straight fin tube model. Therefore, the V-fin tube model should be used for numerical simulation to save the computing resources. These results will make additional contributions to numerical simulation studies in microchannel evaporators, especially with a CO_2_ refrigerant.

## 4. Conclusions

A numerical study of the effect of the fin shape on the heat transfer and the solution time of a microchannel evaporator in a CO_2_ air conditioning system was completed at the evaporative temperature of 10 °C. The two types of fin shapes were mentioned: V-shaped fins and straight fins. The experimental data were used to verify the numerical results. Some important contributions can be expressed as follows:The mathematical model, the boundary conditions, the meshing method and the PARDISO solver were applied to numerically simulate a microchannel evaporator model. The numerical results were in good agreement with those obtained from the experimental results, with an error of less than 10%.For the system under consideration and for the same heat transfer area and the heat transfer coefficient for the air side, the effect of the fin shape on the heat transfer of a microchannel evaporator was not different. However, the solution time and the physical memory for the straight fins were 1.3 and 1.45 times those of the V-fins, respectively. Under the same conditions, the V-fin shape should be used for numerical simulation and not the straight fin shape.The evaporation of the refrigerant in the microchannel evaporator took place in four passes. The normal heat flux from the air through the fins and tubes was almost reached at 1550 W/m^2^ at the evaporative temperature of 10 °C.

## Figures and Tables

**Figure 1 micromachines-13-01648-f001:**
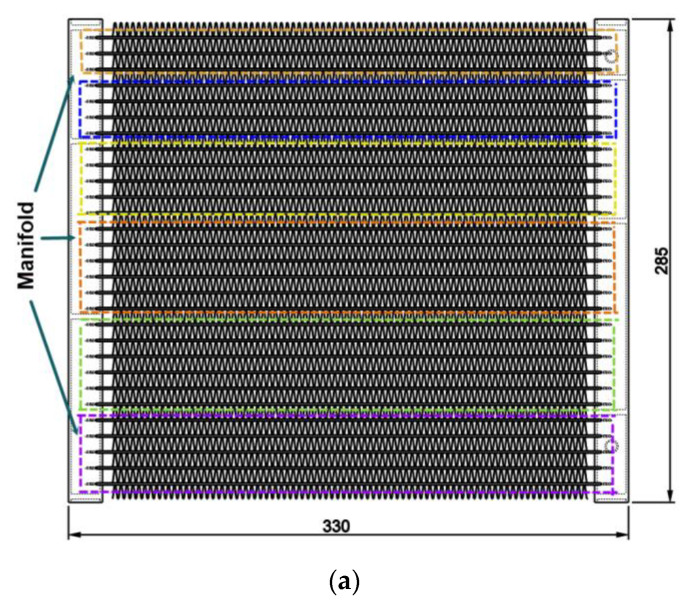
Dimensions of the microchannel evaporator: (**a**) Front view; (**b**) A flat tube drawing with the straight fins; (**c**) A flat tube drawing with the V-fins.

**Figure 2 micromachines-13-01648-f002:**
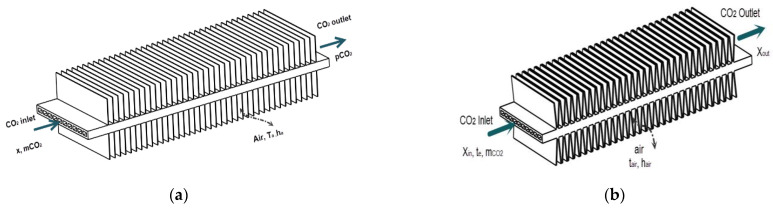
Two flat tube models of the CO_2_ microchannel evaporator: (**a**) a model of straight finned flat tube; (**b**) a model of V-finned flat tube.

**Figure 3 micromachines-13-01648-f003:**
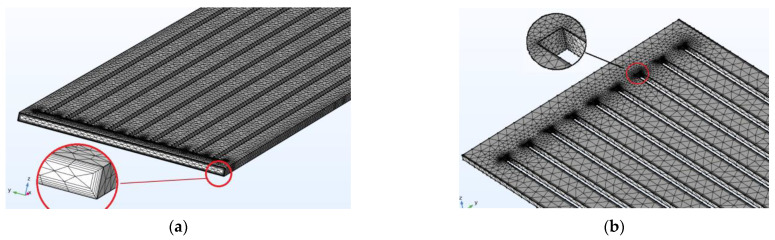
Grid mesh diagram of the straight fin model. (**a**) The boundary layer; (**b**) The edge of the corners.

**Figure 4 micromachines-13-01648-f004:**
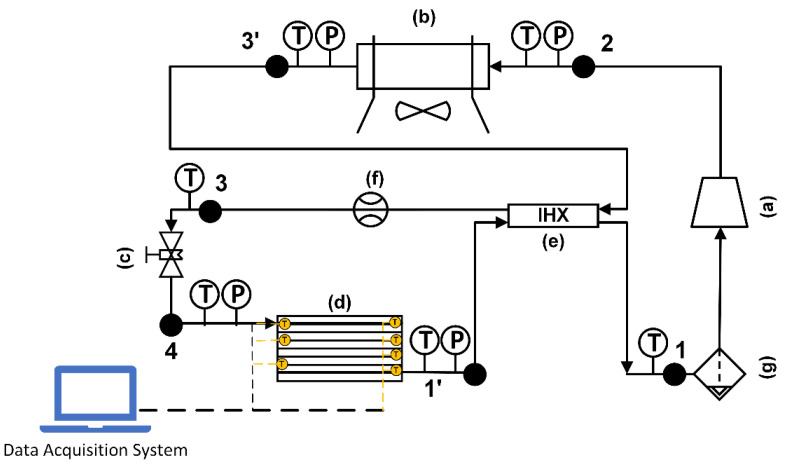
Schematic of the test loop for CO_2_ air conditioning system: (a) a CO_2_ compressor; (b) a gas cooler; (c) a throttle valve; (d) a microchannel evaporator; (e) an internal heat exchanger; (f) a flow meter; (g) a gas–liquid separator.

**Figure 5 micromachines-13-01648-f005:**
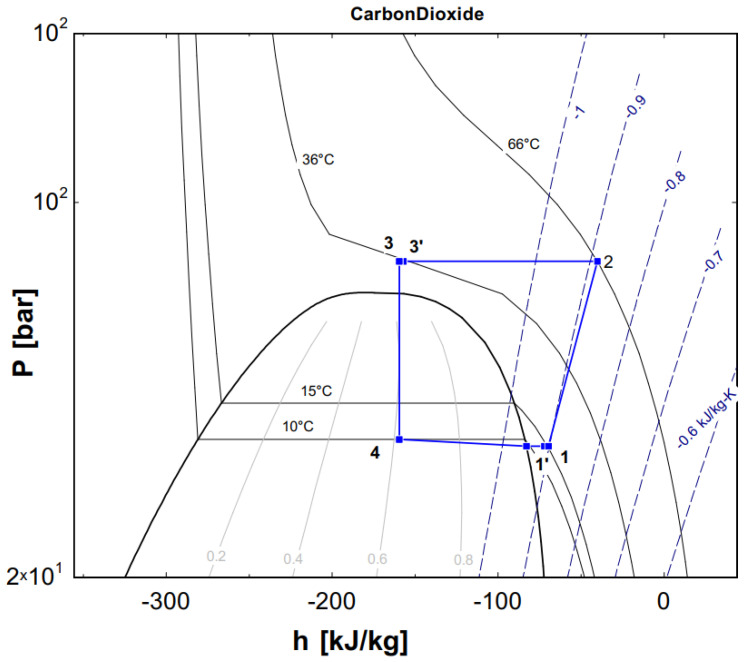
The experimental data presented on a p–h diagram.

**Figure 6 micromachines-13-01648-f006:**
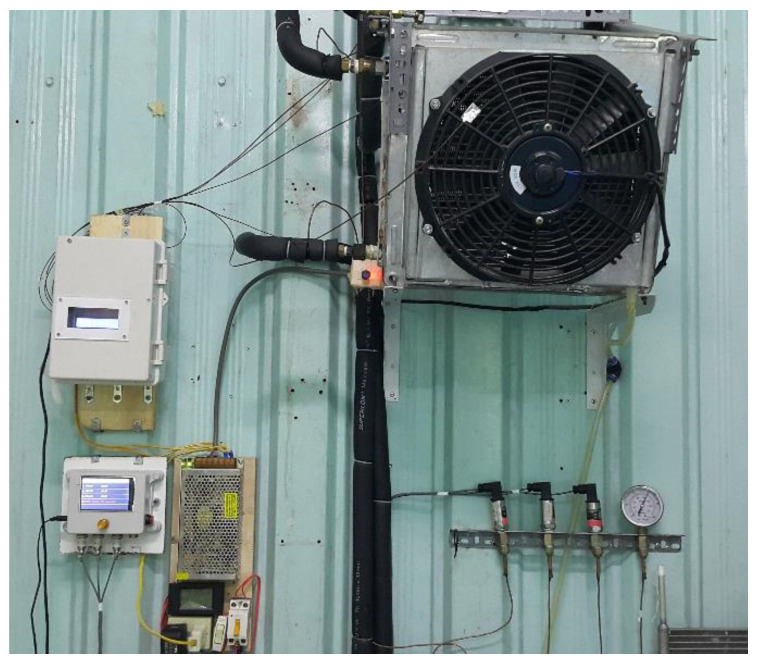
A V-finned tube evaporator for the CO_2_ air conditioning system.

**Figure 7 micromachines-13-01648-f007:**
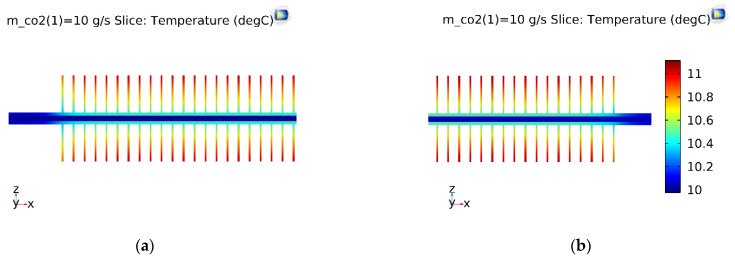
The temperature slice of the straight finned flat tube in the first pass. (**a**) Inlet; (**b**) Outlet.

**Figure 8 micromachines-13-01648-f008:**
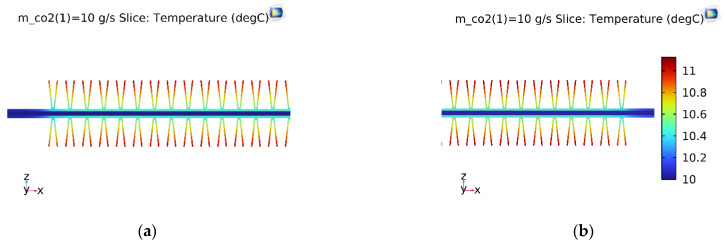
The temperature slice of the V-finned flat tube in the first pass. (**a**) Inlet; (**b**) Outlet.

**Figure 9 micromachines-13-01648-f009:**
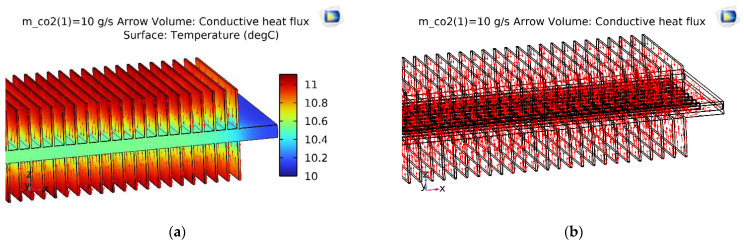
The temperature and conductive heat flux of the straight finned flat tube of the first pass. (**a**) Temperature profile; (**b**) Conductive heat flux.

**Figure 10 micromachines-13-01648-f010:**
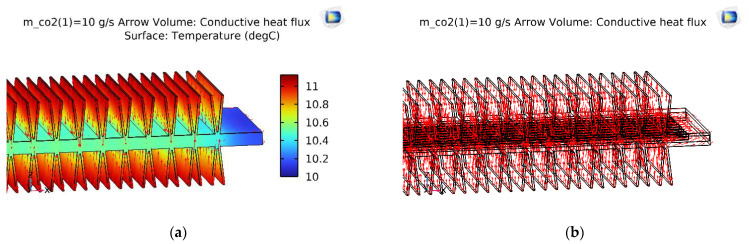
The temperature and conductive heat flux of the V-finned flat tube of the first pass. (**a**) Temperature profile; (**b**) Conductive heat flux.

**Figure 11 micromachines-13-01648-f011:**
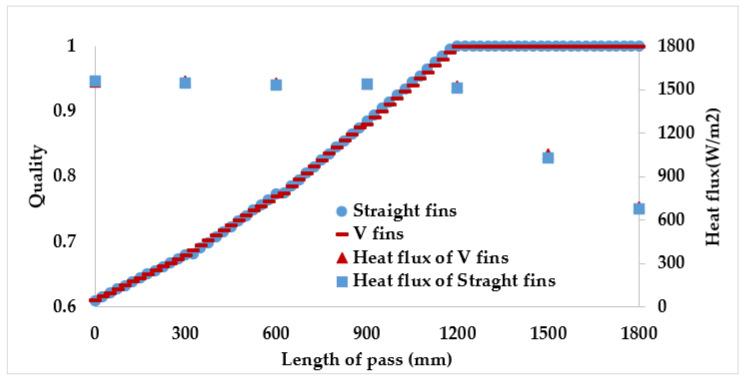
The vapor quality and the heat flux along the length of the flat tube.

**Figure 12 micromachines-13-01648-f012:**
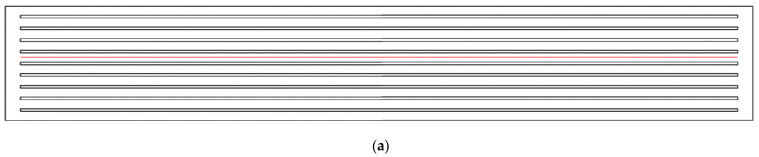
The refrigerant velocity in the microchannel for the straight fin tube model. (**a**) Location of the velocity discussed; (**b**) The velocity field in a microchannel.

**Figure 13 micromachines-13-01648-f013:**
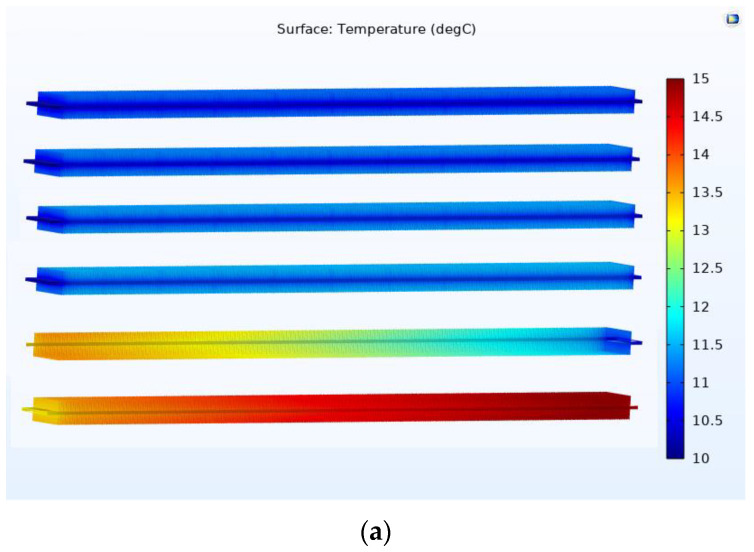
A comparison between the numerical simulation and experimental data. (**a**) The numerical results of the straight finned flat tube model; (**b**) The experimental temperature profile of the evaporator was captured by the thermal camera.

**Table 1 micromachines-13-01648-t001:** Detailed geometries of the microchannel evaporator with the straight fins and the V-fins.

Name	Specifications of Straight Fin	Specifications of V-Fin
Heat transfer area of evaporator (m^2^)	2.5	2.5
Evaporator Size (L × H × W) (mm)	330 × 285 × 16	330 × 285 × 16
Flat tube size (mm)	1.3 × 300 × 16	1.3 × 300 × 16
Number of microchannels in a flat tube	10	10
Microchannel size (mm)	0.6 × 1.2	0.6 × 1.2
Fin size (mm)	4.1 × 0.1 × 16	4.05 × 0.1 × 16
Fin pitch (mm)	1.1	1.1
Angle of fin (deg)	0°	13.8°
Number of fins per flat tube	536	536
Heat transfer area of a flat tube (m^2^)	81.3 × 10^−3^	81.3 × 10^−3^
Number of flat tubes in each pass	3-4-5-6-6-5	3-4-5-6-6-5

**Table 2 micromachines-13-01648-t002:** Inlet parameters and measured parameters.

Pass Number	Inlet Parameter	Outlet Parameter
1	T_1_i_ = 10 °C, p_1_i_ = 45 bar; T_a_ = 25 °C, h_a_ = 110 W/(m^2^K) x_1_i_ = 0.61, m_p_1_ = 30/3 = 10 g/s	T_1_o_, p_1_o_, x_1_o_
2	T_1_o_, p_1_o_ T_a_ = 25 °C, h_a_ = 110 W/(m^2^K) x_2_i_ = x_1_o_, m_p_2_ = 30/4 = 7.5 g/s	T_2_o_, p_2_o_, x_2_o_
3	T_2_o_, p_2_o_ T_a_ = 25 °C, h_a_ = 110 W/(m^2^K) x_3_i_ = x_2_o_, m_p_3_ = 30/5 = 6 g/s	T_3_o_, p_3_o_, x_3_o_
4	T_3_o_, p_3_o_ T_a_ = 25 °C, h_a_ = 110 W/(m^2^K) x_4_i_ = x_3_o_, m_p_3_ = 30/6 = 5 g/s	T_4_o_, p_4_o_, x_4_o_

(where indices 1, 2, 3 and 4 indicate a refrigerant in pass 1, 2, 3 and 4, respectively. Indices i and o indicate an inlet and outlet of the flat tube, respectively. T_a_ is the air room temperature, m is the CO_2_ mass flow rate, x is the vapor quality, and h_a_ is the air convective heat transfer coefficient).

**Table 3 micromachines-13-01648-t003:** Accuracies and ranges of testing devices.

Testing Apparatus	Accuracy	Range
Infrared thermal camera, Fluke Ti9	2%	−20~250 °C
Thermometer, Extech 421,509	0.75% of rdg	−20~250 °C
Thermocouples, T—Type	±0.1 °C	0~100 °C
Digital volumetric flow rate meter	±0.5% FS	400 to 5000 l/h
Pressure sensor, SENSYS—Korea	±0.5 FS	0~100 bar

**Table 4 micromachines-13-01648-t004:** The solution time and the physical memory.

Types	All Domains (Elements)	All Boundaries (Elements)	Solution Time (s)	Physical Memory (GB)
Straight fins	601,833	121,428	1653	9.7
V-fins	618,006	122,592	1268	6.69

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
