# Peer review of "The Effect of Fin Shape on the Heat Transfer and the Solution Time of a Microchannel Evaporator in a CO2 Air Conditioning System—A Numerical Investigation"

_micromachines, 2022, doi:10.3390/mi13101648_

Round 1

Reviewer 1 Report

The manuscript by Nguyen and Dang studies the effect of fin shape on the numerical solution of heat transfer in a microchannel evaporator of a CO2 air conditioning system. Their simulation results are then compared to experiments results. A variation of less than 10% is observed between their simulation and experimental results. The authors also provide the conclusion that simulating a MCHX with a V-fin is less computationally demanding than simulating one with a straight fin even though the results are almost the same for the design under consideration. The results are very interesting to the readers of Microchannels; however, the authors would be required to address my below comments and concerns:  

The manuscript should be revised to improve the quality of English being used for a scientific paper. There are some spelling and grammatical mistakes that should be avoided (e.g. “also affects to their” line 25, etc.)

The sentence in lines 14-15, “So, the V-fin shape should be used to numerically simulate to compare with the straight fin shape.” Is unclear and authors are encouraged to rewrite it to make their point clearer. 

The literature review as provided doesn’t support the authors claim, “From the literature above, it is essential to study the effect of the fin shape on the heat transfer and the solution time of a microchannel evaporator in a CO2 air conditioning system by the numerical method.” The authors are encouraged to rewrite the second and third paragraphs in the introduction (line 36-71) to make it easier for the reader to follow the literature review provided. Specifically, the authors are encouraged to make clear what has been accomplished so far and what was assumed or missing in each of these studies and how that limits the applicability of results provided. The authors would also then need to comment on how their study bridges the gap on what is missing in the literature and why is that important. 

The scale 6.000 in Figure 1b is unclear. The locations of the labels (C) and (D) need to be closer to the views they correspond to in Figure 1b. Also, the scaling ratio on these 2 labels should be written in a way consistent with that used in Figure 1c. 

The length of the heat exchanger is given as 300 mm in Figure 1 and in line 218 but is given as 310 in table 1. The heat transfer area provided in table 1 should be provided in figure 1a. 

Quantity of fins and the angle for the V-fin should be included in table 1.

In equations 12, the letter “P” in “Pk” should be lower case to be consistent with the term used in equations 1, 13 and 14. 

In line 130, how did the authors come up with the “input conditions for the first pass”. Is it based on previous literature or from their own experience.

The values provided in table 2 are unclear. When are the input conditions Tgas cooler =36C; Tsuperheat=10C used? And it seems that x=0.61 is calculated at the outlet of the first pass; whereas, in the results section, the x=0.61 is the inlet to the evaporator. How accurate is Tdischarge=66C? Where can this temperature be found if the outlet temperature from the results seems to be around 15C only. Does the 15C outlet temperature contain the 10C superheat? m=30 g/sec in table 2 wheras m=10 g/sec in figures 6-9. Table 2 should be re-organized by adding the pass number as a column on the very left of the table and providing at least 2 passes, i.e. pass# 1: Inlet Conditions: assumed (based on reference[] or experience), Outlet Conditions: calculated. Pass# 2: Inlet Conditions: same as outlet conditions from pass 1; Outlet Conditions: calculated. 

In figure 4, the direction of the arrow from “g” to “e” should be reversed to go from the IHX to the gas-liquid separator.

In line 188-189, the authors claim that “this result is consistent with the theory of the evaporation.” What about the pressure drop in the microchannels? Is the effect negligible in the first tube?

The authors are encouraged to use consistent figure sizes for parts (a) and (b) in figure 6 and the remaining figures. They are also encouraged to upload higher resolution figures in the final draft. 

In line 218, the authors are encouraged to use an alternative statement to “are not almost different”.

In line 241, was this comparison between simulation and experimental done at an ambient temperature of 33C for both or 33C experimental and 25C for simulation?

The comparison in the temperature profile between simulation and experimental results should be done using the temperatures recorded by calibrated thermocouples since they have better accuracy than the infrared camera. 

Are the authors considering the hottest spot on the figure for the 10% variation in line 246?

The authors are required to comment on what are some of the sources for this variation between simulation and experimental results. Is it possible that their mathematical model underpredicts the pressure drop in the MCHX? What other possible reasons are there?

The second conclusion point is only applicable for the design under consideration and should not be generalized. This point has to be rewritten to make that clear. 

Finally, I recommend the authors to make the article easier follow by the reader by rewriting some of the sections to make the approach clearer.

Author Response

Dear Editor and Reviewer #1:

Thank you for your comments and suggestions on the structure and content of our manuscript. We have revised the manuscript accordingly, and detailed responses to the comments and suggestions are listed below:

1) The manuscript should be revised to improve the quality of English being used for a scientific paper. There are some spelling and grammatical mistakes that should be avoided (e.g. “also affects to their” line 25, etc.)

Ö We appreciate the comments provided by Reviewer #1. Per the review comments, we have revised to improve the quality of English. It is noted that all revisions and additions to the manuscript were marked in red.

2) The sentence in lines 14-15, “So, the V-fin shape should be used to numerically simulate to compare with the straight fin shape.” Is unclear and authors are encouraged to rewrite it to make their point clearer

Ö Thanks for this comment. Following your comment, we have added more information to make this point clearer.

3) The literature review as provided doesn’t support the authors claim, “From the literature above, it is essential to study the effect of the fin shape on the heat transfer and the solution time of a microchannel evaporator in a CO2 air conditioning system by the numerical method.” The authors are encouraged to rewrite the second and third paragraphs in the introduction (line 36-71) to make it easier for the reader to follow the literature review provided. Specifically, the authors are encouraged to make clear what has been accomplished so far and what was assumed or missing in each of these studies and how that limits the applicability of results provided. The authors would also then need to comment on how their study bridges the gap on what is missing in the literature and why is that important.

Ö Per the review comments, we have revised the literature review of the manuscript. It is noted that all revisions and additions to the manuscript were marked in red.

4) The scale 6.000 in Figure 1b is unclear. The locations of the labels (C) and (D) need to be closer to the views they correspond to in Figure 1b. Also, the scaling ratio on these 2 labels should be written in a way consistent with that used in Figure 1c

Ö Thanks for this comment. We have revised the Figure 1.

5) The length of the heat exchanger is given as 300 mm in Figure 1 and in line 218 but is given as 310 in table 1. The heat transfer area provided in table 1 should be provided in figure 1a.

Ö Following your suggestion, we have revised this value in Table 1 and in Figure 1a. The value 300mm is for the length of a flat tube. The value 330mm is for the length of the evaporator.

6) Quantity of fins and the angle for the V-fin should be included in table 1.

Ö Based on your comment, quantity of fins and the angle for the V-fin have added in Figure 1c and in Table 1.

7) In equations 12, the letter “P” in “Pk” should be lower case to be consistent with the term used in equations 1, 13 and 14.

Ö Thanks for this comment. We have revised the letter “pk” in equation 13, 14, 15.

8) In line 130, how did the authors come up with the “input conditions for the first pass”. Is it based on previous literature or from their own experience?

Ö Following your suggestion, we have revised all the values in the Table 2. The input conditions in the first pass are from our experimental data and from the calculation of experimental data.

9) The values provided in table 2 are unclear. When are the input conditions Tgas cooler =36C; Tsuperheat=10C used? And it seems that x=0.61 is calculated at the outlet of the first pass; whereas, in the results section, the x=0.61 is the inlet to the evaporator. How accurate is Tdischarge=66C? Where can this temperature be found if the outlet temperature from the results seems to be around 15C only. Does the 15C outlet temperature contain the 10C superheat? m=30 g/sec in table 2 wheras m=10 g/sec in figures 6-9. Table 2 should be re-organized by adding the pass number as a column on the very left of the table and providing at least 2 passes, i.e. pass# 1: Inlet Conditions: assumed (based on reference[] or experience), Outlet Conditions: calculated. Pass# 2: Inlet Conditions: same as outlet conditions from pass 1; Outlet Conditions: calculated

Ö Following your suggestion, we have re-written all the values in the Table 2. These value (Tgas cooler =36oC, Tsuction=15oC, Tdischarge=66oC) were measured by the thermocouple sensor as shown in Figure 4a. The input vapor quality (x=0.61) can be determined by equation 11 or EES software as shown in Figure 4b.

- The mass flow rate m=30g/s is evaporator input value. In 1st pass, there are 3 flat tubes. So, the mass flow rate for one tube must be divided by 3 equals 10g/s. And so on.

- Table 2 was re-organized by adding the pass number as a column on the very left of the table and providing 4 passes, i.e.: Inlet parameters, outlet parameters. Inlet parameters of pass#2 are the same as outlet parameters from pass#1.

10) In figure 4, the direction of the arrow from “g” to “e” should be reversed to go from the IHX to the gas-liquid separator

Ö Following your suggestion, we have edited this Figure (Figure 4 in the revised manuscript). Furthermore, Figure 4 shows the thermocouple installed at the evaporator.

11) In line 188-189, the authors claim that “this result is consistent with the theory of the evaporation.” What about the pressure drop in the microchannels? Is the effect negligible in the first tube?

Ö Following your deep suggestion, the pressure drop in the first pass is 0.26bar. It decreases when the vapor quality increases. In the flat tube, the pressure drop has little effect on temperature. So, we do not mention it. However, in the experiment, the pressure drop is shown in Figure 4b. (4-1 process).

12) The authors are encouraged to use consistent figure sizes for parts (a) and (b) in figure 6 and the remaining figures. They are also encouraged to upload higher resolution figures in the final draft

Ö Thanks for this comment. We have uploaded higher resolution the Figure 1.

13) In line 218, the authors are encouraged to use an alternative statement to “are not almost different”.

Ö Following your suggestion, we have revised this statement in the manuscript.

14) In line 241, was this comparison between simulation and experimental done at an ambient temperature of 33C for both or 33C experimental and 25C for simulation?

Ö Following your suggestion, 33oC is the temperature of air outside the room. 25oC is the indoor temperature. Both of these temperatures are experimentally measured.

15) The comparison in the temperature profile between simulation and experimental results should be done using the temperatures recorded by calibrated thermocouples since they have better accuracy than the infrared camera.

Ö Following your suggestion, the temperature on the evaporator is measured with both a thermocouple and the infrared camera. Fig. 4 and 5 show 7 thermocouples were installed on the evaporator in the revised manuscript.

16) Are the authors considering the hottest spot on the figure for the 10% variation in line 246?

Ö Thanks for this comment. The deviation between numerical results and experimental data is less than 10% at the hottest spot.

17) The authors are required to comment on what are some of the sources for this variation between simulation and experimental results. Is it possible that their mathematical model underpredicts the pressure drop in the MCHX? What other possible reasons are there?

Ö Thanks for your deep comments. Fig. 4b shows the p-h diagram of experiment. In numerical simulation, the roughness in the microchannel is ignored, so the pressure drop in the MCHX is much smaller than that obtained from the experimental data. The effect of pressure drop in the MCHX will be investigated on a follow-up study.

18) The second conclusion point is only applicable for the design under consideration and should not be generalized. This point has to be rewritten to make that clear.

Ö Per the reviewer’s comments, we have re-written the conclusion of the manuscript.

The revised paper (Second version) is resubmitted to your journal. Again, your efforts in reviewing the manuscript to make it more presentable and in publishing the paper are deeply appreciated.

Best Regards,

Assoc. Prof. Dr. Thanhtrung Dang

Reviewer 2 Report

This study investigated the heat transfer effect of microchannel evaporators with straight fins and V shape fins with numerical analysis and experimental methods. The objective of the study is quite unclear. The numerical methods, governing equations, computational domain and boundary conditions are not clearly presented. No novelty is found in conclusions.

Author Response

Dear Editor and Reviewer #2:

Thank you for your comments and suggestions on content of our manuscript. We have revised the manuscript accordingly, and detailed responses to the comments and suggestions are listed below:

1) The objective of the study is quite unclear.

Ö We appreciate the comments provided by Reviewer #2. Per the review comments, we have revised the literature review and other sections of the manuscript. The objective of the study is clearer in revised manuscript.

2) The numerical methods, governing equations, computation domain and boundary condition are not clearly presented.

Ö Based on your comments, the equation 11 was added; the equations (12 -14) were edited and the Table 2 was re-organized. The numerical methods, governing equations, computation domain and boundary condition were re-presented.

3) No novelty is found in conclusion.

 Ö Thank you for your suggestion, we have revised the conclusions to show novelty. It is noted that all revisions and additions to the manuscript were marked in red

The revised paper (Second version) is resubmitted to your journal. Again, your efforts in reviewing the manuscript to make it more presentable and in publishing the paper are deeply appreciated.

Best Regards,

Assoc. Prof. Dr. Thanhtrung Dang

Reviewer 3 Report

The manuscript deals with The effect of fin shape on the heat transfer of a microchannel evaporator. The core of the study is a numerical simulation on the fin shape of a microchannel evaporator were performed. Two types of fin shapes were dealt with: the straight fins and V-fins. The contents of this paper lays well within the aims and scopes of the Micromachines: Special Issue: Heat and Mass Transfer in Microchannels. Nevertheless, in order to improve the readability and clarity of the manuscript, I suggest a revision. They have to address the following critical aspects before publication:
1. The introduction needs to be better organized in order to show the new contribution with respect to the state of the art. In particular, the Authors should underline the innovation and advance of the proposed contribution. A clear novelty statement could be provided.
2. Uncertainty values for the dependent variables  such as normal heat flux should be given.
3.  More experimental results could be disclosed in a table form.

Author Response

Dear Editor and Reviewer #3:

Thank you for your comments and suggestions on the structure and content of our manuscript. We have revised the manuscript accordingly, and detailed responses to the comments and suggestions are listed below:

1)  The introduction needs to be better organized in order to show the new contribution with respect to the state of the art. In particular, the Authors should underline the innovation and advance of the proposed contribution. A clear novelty statement could be provided

Ö We appreciate the comments provided by Reviewer #3. Per the review comments, we revised the literature review of the manuscript. We also revised the conclusions to show the innovation and advance of the proposed contribution. It is noted that all revisions and additions to the manuscript were marked in red.

2) Uncertainty values for the dependent variables such as normal heat flux should be given.

Per the reviewer’s comments, we added the uncertainty values in the manuscript such as the vapor quality and the normal heat flux.

3) More experimental results could be disclosed in a table form

Ö Thanks for your suggestion, we revised the manuscript to show the experimental results. It is noted that the experimental results in this study are only used to verify the numerical simulation.

The revised paper (Second version) is resubmitted to your journal. Again, your efforts in reviewing the manuscript to make it more presentable and in publishing the paper are deeply appreciated.

Best Regards,

Assoc. Prof. Dr. Thanhtrung Dang

Round 2

Reviewer 1 Report

The authors have improved the manuscript according to initial comments and recommendations provided earlier. The manuscript is now suitable for publication in Micromachines; however, the authors are encouraged to address the following minor recommendations:

1) The authors would need to proofread their manuscript again to eliminate few grammatical mistakes present throughout the different sections.

2) In line 12, the authors should add: “ For the system under consideration, and for” before “the same heat transfer area…”. Since this conclusion can’t be generalized for all microchannel systems. The results are only applicable to this particular system under consideration. The same should be done in line 273 in the conclusion section. 

3) Caption for figure 4 needs to be amended as it’s currently confusing since it only covers subfigure (a) and doesn’t mention anything about subfigure (b). However, it does have (b) as part of subfigure (a).

4) Upload better graph plots in figures 10 and 11 (b) with consistent labels, fonts and background

Author Response

Thank you for your comments and suggestions on the structure and content of our manuscript – Round 2. We have revised the manuscript accordingly, and detailed responses to the comments and suggestions are listed below:

Reviewer #1

1) The authors would need to proofread their manuscript again to eliminate few grammatical mistakes present throughout the different sections.

Ö We appreciate the comments provided by Reviewer #1. Following your comments, we have proofread this manuscript again by a native English - speaking lecturer.

2) In line 12, the authors should add: “For the system under consideration, and for” before “the same heat transfer area …”. Since this conclusion can’t be generalized for all microchannel systems. The results are only applicable to this particular system under consideration. The same should be done in line 273 in the conclusion section.

Ö Thank you for your deep comments, we have added your suggestions in the manuscript to make this point clearer. It is noted that all revisions and additions to the manuscript were marked in red.

3) Caption for figure 4 needs to be amended as it’s currently confusing since it only covers subfigure (a) and doesn’t mention anything about subfigure (b). However, it does have (b) part of subfigure (a).

Ö Per the review comments, we separated Figure 4 into Figure 4 and Figure 5. We have revised the number of Figures.

4) Upload better graph plots in figures 10 and 11 (b) with consistent labels, font and background.

Ö Thanks for this comment. We have uploaded a higher resolution Figure 10 and revised Figure 11 (b).

The revised paper (Third version) is resubmitted to your journal. Again, your efforts in reviewing the manuscript to make it more presentable and in publishing the paper are deeply appreciated.

Best Regards,

Reviewer 3 Report

Accept in present form.